# PR3-ANCAs Detected by Third-Generation ELISA Predicts Severe Disease and Poor Survival in Primary Sclerosing Cholangitis

**DOI:** 10.3390/diagnostics12112682

**Published:** 2022-11-03

**Authors:** Steffi Lopens, Ewa Wunsch, Malgorzata Milkiewicz, Nadja Röber, Grit Zarske, Abdullah Nasser, Karsten Conrad, Martin Laass, Stefan Rödiger, Marcin Krawczyk, Dirk Roggenbuck, Piotr Milkiewicz

**Affiliations:** 1Institute of Biotechnology, Faculty Environment and Natural Sciences, Brandenburg University of Technology Cottbus-Senftenberg, 01968 Senftenberg, Germany; 2Medipan GmbH, 15827 Dahlewitz, Germany; 3Translational Medicine Group, Pomeranian Medical University in Szczecin, 70-204 Szczecin, Poland; 4Department of Medical Biology, Pomeranian Medical University in Szczecin, 70-204 Szczecin, Poland; 5Institute of Immunology, Technical University Dresden, 01307 Dresden, Germany; 6Department of Pediatrics, University Hospital and Medical Faculty Carl Gustav Carus, Technical University Dresden, 01307 Dresden, Germany; 7Faculty of Health Sciences Brandenburg, Brandenburg University of Technology Cottbus—Senftenberg, 01968 Senftenberg, Germany; 8Department of Medicine II, Saarland University Medical Center, 66421 Homburg, Germany; 9Laboratory of Metabolic Liver Diseases, Department of General, Transplant and Liver Surgery, Medical University of Warsaw, 02-097 Warsaw, Poland; 10European Reference Network, 66421 Homburg, Germany; 11European Reference Network, 02-097 Warsaw, Poland

**Keywords:** primary sclerosing cholangitis, antibodies against serine protease proteinase-3, survival, health-related quality of life, liver biochemistry, disease severity

## Abstract

A highly sensitive detection of anti-neutrophil cytoplasmic antibodies to serine proteinase-3 (PR3-ANCAs) aids in the serological diagnosis of autoimmune liver disorders and the prediction of severity in primary sclerosing cholangitis (PSC). Here, we evaluate a novel third-generation ELISA for the detection of PR3-ANCAs. In total, 309 patients with PSC, 51 with primary biliary cholangitis (PBC), and 120 healthy blood donors (BD) were analyzed. For the survival analysis in PSC, the outcome was defined as liver-transplantation-free survival during the follow-up. Positive PR3-ANCA levels were found in 74/309 (24.0%) of patients with PSC. No BDs and one patient with PBC demonstrated PR3-ANCA positivity. PR3-ANCAs were revealed as independent predictors for a poor PSC outcome (study endpoint: liver transplantation/death, log-rank test, *p* = 0.02). PR3-ANCA positivity, lower albumin levels, and higher bilirubin concentrations were independent risks of a poor survival (Cox proportional-hazards regression analysis, *p* < 0.05). The Mayo risk score for PSC was associated with PR3-ANCA positivity (*p* = 0.01) and the disease severity assessed with a model of end-stage liver disease (MELD) and extended MELD-Na (*p* < 0.05). PR3-ANCAs detected by a third-generation ELISA are diagnostic and prognostic markers for PSC. Their wider use could help to identify patients who are at-risk of a more severe disease.

## 1. Introduction

Primary sclerosing cholangitis (PSC) is a chronic immune-mediated, life-threatening, genetically predisposed liver illness with a largely unknown etiopathogenesis [1,2]. The prevalence of PSC is estimated to be up to 16.2 per 100,000 individuals and continues to increase. PSC is characterised by a progressive disease course finally resulting in biliary fibrosis and end-stage liver disease in the majority of cases. PSC confers an increased risk of cholangiocarcinogenesis, independent of the duration and activity of the disease, and is associated with the co-occurrence of inflammatory bowel disease (IBD) in 70% of cases [3]. 

Although PSC has non-typical features of an autoimmune disease, such as unresponsiveness to immunosuppression and a male predominance, it is generally accepted that autoimmunity plays an important role in the etiopathogenesis of PSC [4,5]. Perinuclear anti-neutrophil cytoplasmic antibodies (p-ANCAs), or more precisely peripheral anti-neutrophil nuclear autoantibodies (p-ANNAs), detected by qualitative indirect immunofluorescence have been one of the most debated diagnostic markers for PSC in the past [6]. However, p-ANNAs do not appear to be directly linked to the clinical symptoms of the disease, and despite recent progress in identifying potential targets, their actual targets remain elusive [7,8,9,10]. The involvement of autoimmunity in PSC has been further strengthened by recent data demonstrating that ANCAs to serine proteinase-3 (PR3-ANCAs) detected by bead-based immunoassays can occur in PSC and even have a predictive value for severe disease and the occurrence of cholangiocarcinoma (CCA) [11,12,13]. 

Thus, the occurrence of PR3-ANCAs in up to 54% of PSC patients and their predictive value for disease severity ushered in a new era in the serological diagnosis of PSC as an autoimmune liver disorder [11,14]. Originally, PR3-ANCAs were introduced as serological markers and were just recently confirmed as a classification criterion for granulomatosis with polyangiitis (GPA), an ANCA-associated vasculitis [15,16]. In the context of the serological diagnosis of rapidly progressive glomerulonephritis, PR3-ANCAs are included in the recommended specific autoantibody (autoAb) testing [17,18]. Later, PR3-ANCAs were reported as serological markers for ulcerative colitis (UC), an IBD closely related to PSC, and were suggested to be useful for the differential serological diagnosis of IBD [19,20,21,22,23]. However, in contrast to PSC and UC, PR3-ANCAs did not show an association with disease activity in GPA [24,25]. 

As a result of the complex epitope characteristics of PR3 and the ensuing challenges in autoAb assay development, three generations of immunoassays with a continuous improvement in assay characteristics have been employed for the assessment of PR3-ANCAs, particularly with regard to the serological diagnosis of ANCA-associated vasculitides [26,27]. 

Therefore, we used a novel third-generation enzyme-linked immunosorbent assay (ELISA) for the detection of PR3-ANCAs in a large cohort of PSC patients in comparison with other autoimmune liver illnesses to investigate PR3-ANCA prevalence and association with PSC severity as well as risk scores. Furthermore, the predictive power of PR3-ANCAs for liver-transplantation (LTx)-free survival was investigated.

## 2. Materials and Methods

### 2.1. Patients

We prospectively enrolled 309 Caucasian patients with PSC, 196 (63.4%) of whom were males with a median age of 32 years (interquartile rage [IQR] 29–32 years) at two university medical centres in Poland (Warsaw and Szczecin) (Table 1). The diagnoses of PSC (*n* = 254, 82.2%) or PSC with autoimmune hepatitis (AIH) features (PSC/AIH variant) (*n* = 55, 17.8%) were established using the EASL Guideline [28]. Eighty-two patients had liver cirrhosis confirmed by either histology or imaging techniques (computed tomography or liver elastography)**.** All patients were clinically and endoscopically screened for concomitant IBD: 170 (82.9%) of the PSC patients with concomitant IBD were diagnosed with UC, 12 (5.9%) with unclassified colitis, and 23 (11.2%) with Crohn’s disease. Exclusion criteria were the presence of other chronic liver diseases such as viral hepatitis or alcoholic liver disease. In addition, we excluded patients with comorbidities other than IBD in which ANCAs might be present, such as ANCA-associated vasculitides. We also did not include patients with other severe diseases that could impair quality of life, such as decompensated diabetes mellitus, extrahepatic malignancies, or advanced chronic kidney disease.

As disease controls, 51 Caucasian patients with primary biliary cholangitis (PBC), including 44 (86.3%) females and 7 (13.7%) males from the University Hospital of Dresden, Germany, were included in the study [29]. As expected, women predominated in the PBC cohort in contrast to the PSC cohort (Table 1, *p* < 0.05). Two patients with PBC (3.6%) had liver cirrhosis and 8 (14.5%) had an overlap with AIH (PBC/AIH). Patients with PBC without AIH (64 years, IQR: 58–72 years) and PBC/AIH (58 years, IQR: 56–68 years) had significantly higher median ages at survey than patients with PSC without AIH (32 years, IQR: 26–42 years) and PSC/AIH (26 years, IQR: 21–34 years; *p* < 0.05). One hundred twenty blood donors encompassing 34 (34.0%) females and 86 (86.0%) males with a median age of 33 years (IQR: 25–41 years) were enrolled in the study. Thus, BDs were matched with PSC patients in terms of both age and sex (*p* > 0.05). 

### 2.2. Study Parameters

Blood samples for autoantibody analyses, liver function tests, and clinical parameters were collected at the same appointment (Table A1). All serum samples were aliquoted and stored at −20 °C until serological assessment. Patients were followed for up to 98 months (median follow-up time: 14 months). During the observation period, dates of certain censor events, i.e., LTx or liver-disease-related death as well as the occurrence of CCA, were documented.

### 2.3. Detection of PR3-ANCA

PR3-ANCA IgG levels were ascertained by a novel third-generation ELISA (anti-PR3 hs, GA Generic Assays GmbH, Dahlewitz, Germany) employing purified native PR3 as the solid-phase antigen. In brief, 96-well microtiter plates (Thermo Fisher Scientific, Schwerte, Germany) were coated with a bridging molecule, followed by incubation with human PR3 from neutrophils (AROTEC Diagnostics ltd, Wellington, New Zeeland) in bicarbonate buffer (pH 9.5) for 24 h at 4 °C. After blocking with 0.05 mol/L Tris/HCl, 1% bovine serum albumin (Tris-BSA, pH 7.4) at room temperature (RT) for one hour, serum samples diluted to 1:100 in dilution buffer (TrisBSA) were incubated at RT for one hour and subsequently washed. Horseradish-peroxidase-conjugated antihuman IgG (Seramun, Heidesee, Germany) was added and developed with a ready-to-use H_2_O_2_/tetramethylbenzidine substrate (Seramun Diagnostica GmbH, Heidesee, Germany) after incubation for 30 min. The substrate reaction was stopped with 0.25 mol/L sulphuric acid after 15 min. The optical density (OD) of the samples was read using a microplate reader (SLT, Crailsheim, Germany) at a wavelength of 450/620 nm. PR3-ANCA concentrations of the samples were directly read in units per millilitre against the respective OD values using a standard curve with 5 calibrators. Samples with values ≥ 15 U/mL were scored positive. The intra-assay precision was 5.5% and 5.8% for samples with PR3-ANCA values of 176 and 6 U/mL, respectively, whereas the inter-assay precision was 9.1% and 13.1% for samples with 126.9 and 18.3 U/mL, respectively.

### 2.4. Health-Related Quality of Life

The Medical Outcomes Study 36-Item Short Form Health Survey (SF-36) and PBC-40 were used to investigate the relationship between PR3-ANCA levels and health-related quality of life (HRQoL) [30,31]. For the use of the SF-36 v.1 questionnaire, a license was obtained in this study (licence number QM044529). Of note, the applicability of PBC-40 for patients with PSC encompassing fatigue, cognitive and social-emotional symptoms, itch, and other symptoms was recently shown [32]. 

### 2.5. Statistics

The Shapiro–Wilk normality test was used to examine the distribution of quantitative variables reported as median values with corresponding interquartile ranges (IQR) where appropriate. Categorical data were described using the number of observations and absolute frequencies. The follow-up time was the time until death, liver transplantation, or the last contact with the patient. A receiver-operating characteristic curve analysis was used to determine performance data such as sensitivity and specificity. The Kruskal–Wallis test with post hoc analysis according to Conover was applied to calculate the differences between subgroups. A correlation analysis was performed using the Spearman rank correlation method. The prevalence comparison between groups was performed by the two-tailed Fisher’s exact test. The survival analysis was performed using the Kaplan–Meyer analysis with a log-rank test. To identify independent variables for the risk prediction of censored events (i.e., LTx or liver-disease-related death) and the occurrence of CCA, Cox proportional-hazards regression was performed including parameters with significant correlations (*p* < 0.05) in the univariate analysis as covariates with regard to clinical and laboratory parameters. Calculations and graphs were performed using MedCalc (MedCalc Statistical Software version 14.8.1, MedCalc Software bvba, Ostend, Belgium). A *p* value < 0.05 was considered statistically significant.

## 3. Results

### 3.1. The Prevalence of PR3-ANCA in Patients and Controls

The novel PR3-ANCA ELISA showed positive PR3-ANCA levels in 74 (24.0%) of 309 patients with PSC using the cut-off of 15 U/mL (Figure 1). No positive PR3-ANCA levels were detected in 120 age- and gender-matched blood donors using the recommended cut-off of 15 U/mL. Of 51 patients with PBC, 1 (2.0%) patient demonstrated positive PR3-ANCA levels. These data resulted in a specificity of 99.4% (95% confidence interval [CI]: 96.8–100.0%) (Table 2).

PR3-ANCA levels in patients with PSC were significantly higher compared with the levels of PBC patients and BDs (Kruskal–Wallis test with Conover’s post hoc analysis, *p* < 0.05) (Figure 1). The receiver-operating characteristic curve analysis revealed an area under the curve of 0.727 (95% CI: 0.683–0.772, *p* < 0.0001) (Figure A1). Using a lower cut-off of 7.5 U/mL as similarly suggested for CLIA PR3-ANCA assays, a sensitivity of 41.8% with a specificity of 94.2% was obtained (Table 2). At this cut-off, eight (15.7%) patients with PBC and two (1.7%) BDs showed positive PR3-ANCA levels.

### 3.2. Correlation of PR3-ANCA Positivity with PSC Characteristics and Phenotypes

PR3-ANCA positivity was significantly more prevalent in PSC patients with concurrent IBD (OR = 1.9, 95% CI: = 1.1 to 3.6; *p* = 0.027), especially in those with UC (OR = 2.3, 95% CI: = 1.3 to 4.1; *p* = 0.003) (Table 3). Of 170 PSC patients with UC, 52 (30.6%) patients demonstrated positive PR3-ANCA levels. There was no significant difference between PSC patients with and without CD (*p* > 0.05).

### 3.3. Correlation of PR3-ANCA Positivity with Laboratory Parameters and Health-Related Quality of Life

PR3-ANCA positivity was significantly more prevalent in PSC patients with poorer liver parameters (Table 4). Patients suffering from PSC with positive PR3-ANCA levels showed significantly higher liver enzyme levels encompassing liver transaminases (aspartate aminotransferase, alanine aminotransferase), alkaline phosphatase, and gamma glutamyltransferase (*p* < 0.05). Furthermore, PR3-ANCA-positive patients demonstrated higher bilirubin levels (*p* < 0.01). In contrast, there was no correlation between PR3-ANCA levels and HRQoL obtained by mental and physical component summary scoring systems (*p* > 0.05). However, PSC patients with far out values (Figure 1) were younger at diagnosis and study enrolment (24 years vs. 29 years, 26 years vs. 32 years, *p* < 0.05, respectively).

### 3.4. Association of PR3-ANCA Positivity with End-Stage Liver Disease and PSC Risk Scores

PSC severity assessed by the model for end-stage liver disease (MELD) and extended MELD-Na was significantly associated with PR3-ANCA positivity in PSC patients (Table 4, *p* < 0.05). Furthermore, the Mayo risk score for PSC was significantly associated with PR3-ANCA positivity (*p* = 0.01).

PR3-ANCA levels were significantly correlated with MELD and extended MELD-Na (Spearman’s rho 0.194, 95% CI: 0.082–0.301, *p* < 0.001; 0.186, 95% CI: 0.074–0.294, *p* = 0.001, respectively).

### 3.5. PR3-ANCA Occurrence Impacts Liver-Transplant-Free Survival in PSC

During the follow-up time (median: 14 months), a total of 80 (25.9%) patients with PSC reached a study-defined endpoint. Seventy-three (23.6%) patients were transplanted and seven (2.3%) died. The event-free survival rate in the whole study cohort was 74.1%. The log-rank test yielded a hazard ratio of 1.8 (95% confidence interval (CI): 1.1 to 3.2) for PR3-ANCA positivity, and the Kaplan–Meier survival analysis clearly demonstrated a significant association of PR3-ANCA positivity with a shorter LTx-free survival (*p* = 0.03) (Figure 2). To identify independent variables for the risk prediction of the censored events, we first performed a univariate regression analysis followed by a multivariate Cox hazards regression analysis. We used the independent clinical and demographic parameters with a significant association in univariate analyses for the multivariate analysis (Table A2). As a result, PSC cases with PR3-ANCA positivity, lower albumin levels, and higher bilirubin concentrations were at risk of poor survival (Table 5).

### 3.6. Association between PR3-ANCA Positivity and the Risk of CCA

During the follow-up, CCA was diagnosed in 13 (4.2%) of the 309 patients. Among these patients, four (30.8%) showed PR3-ANCA positivity at a cut-off of 15 U/mL. Using a lower cut-off of 6 U/mL, eight (61.5%) PSC patients with concurrent CCA showed a positive result with the novel third-generation ELISA. There was a tendency for an association of PR3-ANCA positivity at this cut-off with the occurrence of CCA in this PSC cohort (log-rank test, *p* = 0.16).

## 4. Discussion

The reported association of PR3-ANCAs with disease severity in patients with UC and, more recently, PSC, as well as their prognostic value for the latter has prompted debate over the disease specificity of PR3-ANCAs and the methods used for their analysis [11,21]. To date, only bead-based PR3-ANCA assays employing chemiluminescence as a read-out method and characterised by high sensitivity have demonstrated a significant association with disease activity and even prognostic features for LTx-free survival and the occurrence of CCA in PSC [11,12]. Here, we report the association of PR3-ANCAs detected by a novel third-generation ELISA with disease severity in PSC with a prospective study. Our data corroborated a PSC specificity of PR3-ANCAs recently shown by a third-generation PR3-ANCA ELISA and bead-based assays [12,13,19]. Moreover, we report a significant association of PR3-ANCAs with the established Mayo risk score for PSC for the first time. 

Until recently, PSC-specific autoAbs, such as perinuclear or atypical ANCAs detected by indirect immunofluorescence and interacting with yet-undefined targets, have not shown a correlation with the severity and prognosis of PSC [33]. Thus, the recent evidence of PR3-ANCAs along with IgA against glycoprotein-2 as prognostic parameters for PSC severity and cholangiocarcinogenesis has called the disease specificity of PR3-ANCAs into question [11]. In fact, PR3-ANCAs were originally established as diagnostic markers for ANCA-associated vasculitis, which was included in the recently refined classification criteria for GPA [15]. This widely accepted concept of the use of PR3-ANCAs has even led to the delayed diagnosis of PSC in PR3-ANCA-positive patients due to the originally suspected and later unconfirmed diagnosis of hepatically localised GPA [34]. 

Third-generation PR3-ANCAs using anchor molecules for the appropriate solid-phase immobilisation of PR3 as an autoantigenic target have been successfully used with improved assay performance, especially in terms of sensitivity in the serology of autoimmune vasculitides [26]. In contrast to previous assay generations, these ELISAs appear to better provide GPA-specific epitopes for sensitive PR3-ANCA binding [27]. 

In our study, up to a quarter of patients in the large PSC cohort studied had positive PR3-ANCA levels using the new third-generation ELISA with the recommended cut-off. This prevalence of PR3-ANCAs was lower compared with the prevalence obtained with other third-generation ELISAs and bead-based assays in other studies demonstrating a prevalence of up to 53.8% [11,12,13]. However, using a lower cut-off of 7.5 U/mL as suggested for chemiluminescent assays (10 U/mL instead of 20 U/mL) increased the sensitivity of PR3-ANCA measurement by the third-generation ELISA to 41.8%, which was not significantly different to values reported with other assay techniques. However, the lowered cut-off decreased the specificity from 99.4% to 94.2%, resulting in 8 of 51 (15.7%) PR3-ANCA-positive patients with PBC. Thus, depending on the clinical situation and an established PSC diagnosis, the lower cut-off of 7.5 U/mL may contribute to identifying more PSC patients at risk of severe disease. 

The revealed PR3-ANCA positivity in patients with PSC by the novel third-generation PR3-ANCA ELISA in our study corroborated recent data that suggested the occurred autoAbs recognised either a similar epitope set characteristic for ANCA-associated vasculitides or new epitopes only characteristic for the loss of tolerance in PSC or a combination of both. The prospective nature of our study with respect to the PSC cohort allowed us to establish a prognostic value of the loss of PR3 tolerance for worse outcomes in PSC, as was also recently shown elsewhere [11]. This supported the assumption that neutrophils are involved in the pathophysiology of PSC. Neutrophils, as innate immune cells that can interact with the microbiota, infiltrate bile ducts of patients with PSC [35]. The occurrence of PR3-ANCAs in PSC could be the result of an impaired interaction of neutrophils with the biliary or intestinal microbiota. The correlation of PR3-ANCAs with the established Mayo risk score for PSC reported in our study for the first time was consistent with the assumption that the loss of tolerance against these cellular components of the innate immunity may play a pathophysiological role at least in patients with severe PSC. Such a correlation has yet only been demonstrated for IgA to glycoprotein-2, the other autoAb predictive for PSC severity [11,36]. In fact, suggested models for risk stratification in PSC, such as the PREsTo and the Amsterdam–Oxford model, have not yet employed autoAbs as serological markers for PSC [36,37]. Thus, both PR3-ANCAs and autoAbs against glycoprotein-2 are regarded as an interesting option in this respect. 

In our study, we confirmed the fact that PR3-ANCAs are not linked with patients’ quality of life, which is one of the most valid aspects of disease severity from a patient’s perspective [11]. Thus, unlike antimicrobial glycoprotein-2 (GP2), the loss of tolerance to neutrophil targets did not appear to be associated with quality of life. Therefore, autoAbs to the large isoform of GP2 remain the only autoAbs to date that are linked with HRQoL and enable to target patient populations with a special need of attention. Furthermore, this finding appeared to highlight the role of immunomodulators such as GP2 in the pathophysiology of PSC.

Our data showed that the presence of PR3-ANCAs detected by the third-generation ELISA was an independent predictor of liver-disease-related death or LTx. The multivariate Cox regression analysis indicated that PR3-ANCA positivity in patients with lower albumin and higher bilirubin levels predicted death or LTx in PSC. This finding corroborated the predictive power of PR3-ANCAs detected with chemiluminescent assays in combination with other parameters, such as lower albumin levels.

However, we could not corroborate the recently reported association of PR3-ANCAs detected with chemiluminescent assays with cholangiocarcinogenesis in PSC [11]. There was only a tendency for PR3-ANCA positivity at a lower cut-off of 6 U/mL with CCA in the PSC cohort. One of the reasons for this could have been the low occurrence of CCA of 4.2% (13/309) in our extensive PSC cohort.

Given the search for prognostic models for risk stratification in PSC, our data supported the inclusion of autoAbs such as PR3-ANCAs and IgA against glycoprotein-2 as autoimmune markers for PSC. In terms of PR3-ANCAs, third-generation ELISAs with appropriate assay performance can be used in this context for new models, which would take into account antibody status as a novel predictive score for severe disease course. 

We acknowledge some limitations of our study. Despite our complex observations encompassing demographic, clinical and laboratory parameters, as well as patients’ measures of disease-related symptoms, patient cohorts enrolled in the study were not completely matched in terms of age and gender. Furthermore, no data on the composition of the microbiota correlating with PR3-ANCA levels were obtained. 

In summary, our findings confirmed the high prevalence of PR3-ANCAs in patients with PSC by using a third-generation ELISA. Moreover, PR3-ANCAs identified patient populations, who had a distinct, more severe disease phenotype with a shorter, non-cancer-related survival. We conclude that the current study opens new avenues to construct and evaluate a model that can take into account antibody status as a novel predictive score for severe disease course, poor survival, and finally the development of CCA.

## Figures and Tables

**Figure 1 diagnostics-12-02682-f001:**
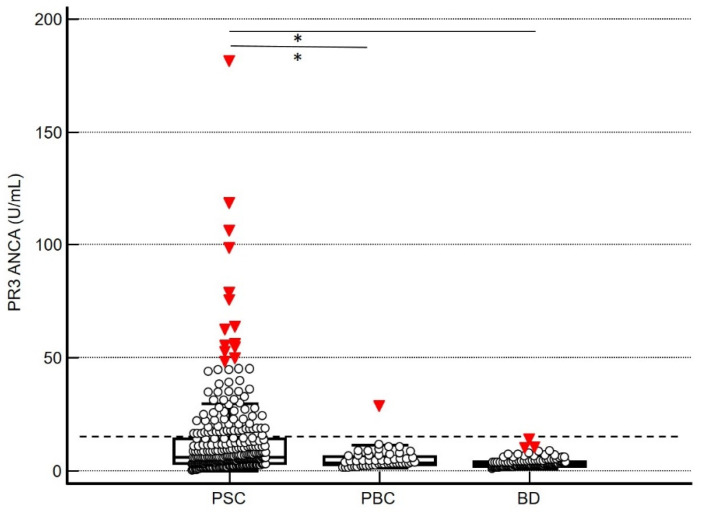
PR3-ANCA levels by third-generation ELISA in sera of 309 patients with primary sclerosing cholangitis (PSC), 51 patients with primary biliary cholangitis (PBC) and 120 blood donors. Data are displayed as units per milliliter in box-and-whisker plots with far out values defined as values that are smaller than the lower quartile minus three times the interquartile range or larger than the upper quartile plus three times the interquartile range, displayed as solid red triangles. The recommended cut-off of 15 U/mL is indicated as a horizontal dashed black line. * *p* < 0.05, Kruskal–Wallis test with Conover’s post hoc analysis.

**Figure 2 diagnostics-12-02682-f002:**
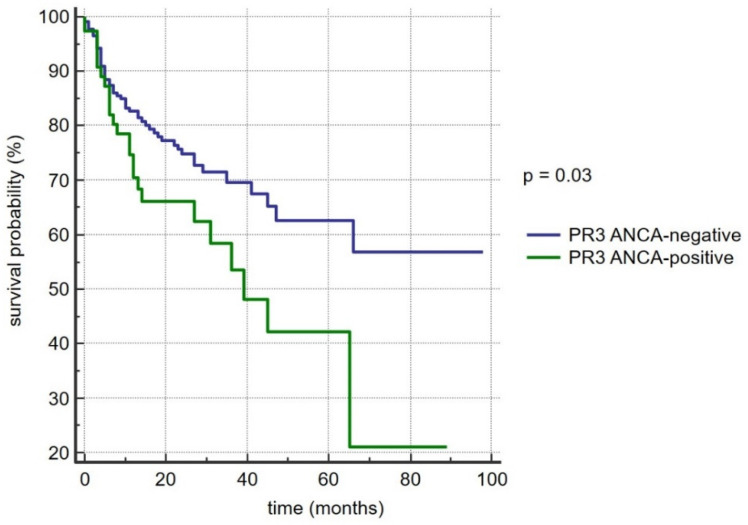
Kaplan–Meier survival curve for 309 patients with primary sclerosing cholangitis, depending on PR3-ANCA status.

**Table 1 diagnostics-12-02682-t001:** Demographic and clinical data of the study cohorts. Abbreviations: autoimmune hepatitis, AIH; blood donors, BDs; female, F; inflammatory bowel disease, IBD; primary biliary cholangitis, PBC; male, M; primary sclerosing cholangitis, PSC; years, y.

	PSC (*n* = 309)	PBC (*n* = 51)	BDs (*n* = 120)
age (IQR), y	32 (29–39)	64 (57–70)	33 (25–41)
age at diagnosis (IQR), y	28 (21–36)	nd	na
disease duration (IQR), y	1 (0–5)	nd	na
gender (M/F)	196/113	7/44	86/34
IBD (yes/no)	205/104	0/51	0/120
IBD subtype			
ulcerative colitis	170	0	0
Crohn’s disease	23	0	0
IBD unclassified	12	0	0
overlap with AIH (yes/no)	55/254	8/43	na
cirrhosis (yes/no)	82/224 *	2/49	na

* No data for three (1%) patients.

**Table 2 diagnostics-12-02682-t002:** Assay performance analysis of PR3-ANCA measurements by the third-generation ELISA using the recommended cut-off of 15 U/mL and a lowered cut-off of 7.5 U/mL. Abbreviations: confidential interval, CI; negative predictive value, NPV; positive predictive value, PPV; positive likelihood ratio, +LR; negative likelihood ratio, −LR.

Parameter	Cut-Off 15 U/mL	95% CI	Cut-Off 7.5 U/mL	95% CI
sensitivity	24.0%	19.3–29.1%	41.8%	36.2–47.5%
specificity	99.4%	96.8–100.0%	94.2%	89.5–97.2%
+LR	41.0	5.78–292.0	7.1	3.9–13.2
−LR	0.8	0.7–0.8	0.6	0.6–0.7
PPV	98.7%	91.2–99.8%	92.8%	87.5–95.9%
NPP	42.0%	40.4–53.5%	47.2%	44.7–49.8%

**Table 3 diagnostics-12-02682-t003:** Association of PR3-ANCA positivity with characteristics of 309 patients with primary sclerosing cholangitis. *Abbreviations:* autoimmune hepatitis, AIH; inflammatory bowel disease, IBD; Crohn’s disease, CD; ulcerative colitis, UC; years, y.

	PR3-ANCA		
	Negative (*n* = 235)	Positive (*n* = 74)	*p* Value ^§^
age at diagnosis (y)	29 (21–38)	27 (20–34)	0.29
age at survey (y)	32 (26–42)	30 (25–36)	0.13
PSC duration (y)	1 (0–5)	1 (0–6)	0.62
gender (male/female)	143 (60.9%)/92 (39.1%)	53 (71.6%)/21 (28.4%)	0.10
overlap with AIH (yes/no)	41 (17.4%)/194 (82.6%)	14 (18.9%)/60 (81.1%)	0.86
cirrhosis (yes/no) *	60 (25.5%)/175 (74.5%)	22 (31.0%)/49 (69.0%)	0.36
IBD (yes/no)	148 (63.0%)/87 (37.0%)	57 (77.0%)/17 (23.0%)	0.03
UC (yes/no)	118 (50.2%)/117 (49.8%)	52 (70.3%)/22 (29.7%)	<0.01
CD (yes/no)	18 (7.7%)/217 (92.3%)	5 (6.8%)/69 (93.2%)	1.00

* Data not available for three patients; ^§^ Mann–Whitney and Fisher’s exact test; *p* value < 0.5 is considered significant.

**Table 4 diagnostics-12-02682-t004:** Association of PR3-ANCA positivity with laboratory parameters and risk scores in 309 patients with primary sclerosing cholangitis. Data presented as median with interquartile range. Abbreviations: alanine aminotransferase, ALT; alkaline phosphatase, ALP; aspartate aminotransferase, AST; gamma glutamyltransferase, GGT; primary biliary cholangitis disease specific quality of life measure, PBC-40; international normalised ratio for prothrombin time, PT-INR; model for end-stage liver disease, MELD; primary sclerosing cholangitis, PSC.

		PR3-ANCA	
	Negative (*n* = 235)	Positive (*n* = 74)	*p* Value
heamoglobin (g/dL)	14 (12–15)	13 (12–14)	0.94
platelets (tys/uL)	242 (160–301)	245 (163–310)	0.94
bilirubin (mg/dL)	0.9 (0.5–2.1)	1.4 (0.7–2.3)	<0.01
ALP (IU/L)	225 (126–377)	363 (169–548)	<0.001
GGT (IU/L)	169 (86–330)	244 (114–496)	<0.01
ALT (IU/L)	70 (40–123)	92 (54–178)	<0.01
AST (IU/L)	58 (33–99)	78 (40–112)	0.02
albumin (g/dL)	4.1 (3.8–4.5)	4.1 (3.6–4.4)	0.13
PT-INR	1 (1.0–1.1)	1.1 (1.0–1.2)	0.09
Na (mmol/L)	140 (139–142)	140 (138–141)	0.06
PBC-40 cognitive	7 (6–15)	11 (8–15)	0.54
PBC-40 fatigue	26 (18–34)	22 (15–31)	0.08
PBC-40 itch	4 (3–7)	4 (3–8)	0.47
PBC-40 other symptoms	14 (10–18)	13 (10–16)	0.15
PBC-40 social	30 (23–41)	30 (25–36)	0.63
PBC-40 symptoms	7 (4–9)	6 (5–8)	0.14
MELD (points)	7 (6–10)	8 (7–11)	0.02
MELD-Na (points)	8 (7–11)	9 (7–13)	<0.01
Mayo risk score for PSC (points)	−0.52 (−1.16–0.49)	−0.21 (−0.66–0.91)	0.01

*p* value < 0.5 is considered significant.

**Table 5 diagnostics-12-02682-t005:** Cox proportional-hazards regression of independent variables for the risk prediction of poor survival (liver transplantation or death) in 309 patients with primary sclerosing cholangitis. Abbreviations: confidence interval, CI; Exp (b), relative risk of event; liver transplantation, LT.

Dependent Variable	Independent Variable	Exp (b)	95% CI Exp (b)	*p* Value
poor survival (death/LT)	PR3-ANCA	2.0169	1.2118 to 3.3570	0.007
	albumin	0.1727	0.1147 to 0.2600	<0.0001
	bilirubin	1.0923	1.0585 to 1.1273	<0.0001

## Data Availability

Not applicable.

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
