# Peer review of "PR3-ANCAs Detected by Third-Generation ELISA Predicts Severe Disease and Poor Survival in Primary Sclerosing Cholangitis"

_diagnostics, 2022, doi:10.3390/diagnostics12112682_

Round 1

Reviewer 1 Report

In this study, the Authors aimed to assess a novel third-generation ELISA for detecting anti-neutrophil cytoplasmic antibodies IgG to serine proteinase 3 (PR3-ANCA) in predicting severity in primary sclerosing cholangitis (PSC). 

According to literature data and current international guidelines, PSC patients classically develop the so-called atypical ANCA (pANNA). PR3-ANCA are well-known serological markers of a subgroup of vasculitides (Wegener's granulomatosis). In the introduction, the authors support the rationale of the study with references (6-7) that refer to their previous studies. However, no other studies confirming similar findings are reported.

Author Response

We thank this Reviewers for her/his helpful comments. We would like to reply point by point as follows:

Reply 1.1: With all due respect, we would like to disagree with the opinion of Reviewer 1. The occurrence of PR3-ANCA in patients with inflammatory bowel disease, particularly ulcerative colitis, without concomitant autoimmune vasculitides has been known since 2013 (Arias-Loste et al, 2013, reference [REF] 14). Several independent groups and we have confirmed these data (Sowa et al, 2014, REF 8; Horn et al. 2018, REF 15; Aoyama et al., 2021, REF16). Patients with ulcerative colitis often have concurrent liver autoimmunity in the form of primary sclerosing cholangitis. Not surprisingly, an independent group in 2014 demonstrated PR3 ANCA positivity in this patient cohort using chemiluminescence as the assay method (Stinton et al., 2014, REF 7). We confirmed these data by another assay technique (microbead-based multiplex immunofluorescence) and chemiluminescence assay (Sowa et al., 2014, REF 8; Wunsch et al., 2021, REF). Therefore, the conclusion of this Reviewer that no other independent studies using other assays support our data is not valid and must be rejected.

We agree upon the notion that patients with ulcerative colitis and primary sclerosing cholangitis demonstrate atypical ANCA or more precisely peripheral anti-neutrophil nuclear autoantibodies (p-ANNA). These autoantibodies detected by qualitative indirect immunofluorescence have been one of the most debated diagnostic markers in primary sclerosing cholangitis in the past (Terjung and Spengler, 2005). After the promising discovery of a neutrophil, nuclear envelope-target molecule for p-ANNA, Terjung et al. identified beta-tubulin isotype 5 as a novel ANCA autoantigen in PSC (Terjung et al., 2010). This target shares a high structural homology with the microbial cell division protein FtsZ. Unfortunately; this finding could not be corroborated in other studies (deBeeck et al., 2011). Moreover, these autoantibodies as well as p-ANNA do not seem to be directly associated with the clinical symptoms of the illness. Thus, the actual targets of p-ANNA remain elusive, and a primary sclerosing cholangitis-specific epitope structure of PR3 could be a candidate (Lopens et al., 2020).

This Reviewer did not offer any critique regarding the study design, data evaluation or conclusions. 

Reviewer 2 Report

Reviewer’s comments

This original article primarily focused on the efficacy of PR3-ANCA in patients with PSC. This article has its own originality and seems to be very impressive. However, the experimental design is partially inappropriate. The interpretations of results obtained from this study seem to be insufficient. It is regrettable to say that this article is not acceptable for publication. Please refer to the comments shown below.

Major

#1. “Disease severity” should be also selected as one of the keywords in this article.

#2. The statement on informed consent is missing in the text. Exclusion criteria should be also mentioned.

#3. The authors should note when PR3-ANCAs were assayed in the enrolled patients (at entry? Or before treatment?). Were the titers of PR3-ANCA correlated with disease severity such as MELD score? In addition, did the authors confirm longitudinal change in the titers of PR3-ANCA in each individual? Was a significant decrease in the titer found after liver transplantation?

#4. Previous studies that p-ANCA was commonly detected in patients with PSC should be cited in “Introduction”. It has been well established that PR3 is a main target antigen of c-ANCA. Did authors check immunofluorescent patterns of ANCA by the indirect immunofluorescence (p-ANCA or c-ANCA) in the enrolled patients seropositive for PR3-ANCA?

#5. The result of the correlation between PR3-ANCA positivity and health-related QOL should be addressed in Table 4, although its statement was found in the text (lines 218-220). Moreover, the authors should describe or speculate the reason why PR3-positivity was not correlated with the QOL in patients with PSC, although it was significantly correlated with disease severity.

#6. Solid triangles indicated individuals with high titer od PR3-ANCA in Figure 1. However, the reason why such patients were listed up from three categories (PSC, PBC and BD groups) remains uncertain. The authors should explain the reason for listing up such patients. Are there any clinical characteristics in such patients? 

#7. The lower cut-off value (7.5U/ml) for PR3-ANCA was also set for the distinction of PSC patients from other patients. The setting of lower cut-off value led to the increase in sensitivity and decrease in specificity. Which is better as the cut-off value for PR3-ANCA, original one (15 U/ml) or lower one (7.5 U/ml)? The authors should answer the question.

#8. Positive and negative predictive values (PPV and NPV) for PSC by PR3-ANCA detection should be also noted in Table 2.

#9. The authors should describe the subtype of immunoglobulin in PR3-ANCA positive patients.(IgA dominant? Or IgG dominant?) In addition, what did AIH features mean in Table 3? The authors should clearly describe the definition of autoimmune features. Serum IgG levels and the prevalence of ANA should be determined in PSC patients.

Minor

#1. Table 2 (line 231) should be corrected to Table 4.

#2. INR (line 226) should be corrected to PT-INR.

Author Response

We thank this Reviewer for her/his positive opinion that the manuscript is original and impressive.

2.1. “Disease severity” should be also selected as one of the keywords in this article.

Reply 2.1: We added the term disease severity as keyword.

2.2. The statement on informed consent is missing in the text. Exclusion criteria should be also mentioned.

Reply 2.2: The statement on informed consent has been given on page 10 line 362.

We added the following exclusion criteria in the Materials and Methods/Patients section:

“Exclusion criteria were the presence of other chronic liver diseases such as viral hepatitis or alcoholic liver disease. In addition, we excluded patients with comorbidities other than IBD in which ANCA might be present, such as ANCA-associated vasculitides. We also did not include patents with other severe diseases, which could impair quality of live such as decompensated diabetes mellitus, extrahepatic malignancies or advanced chronic kidney disease. We also did not include patents with other severe diseases which could impair quality of live such as decompensated diabetes mellitus, extrahepatic malignancies or advanced chronic kidney disease.”

2.3. The authors should note when PR3-ANCAs were assayed in the enrolled patients (at entry? Or before treatment?). Were the titers of PR3-ANCA correlated with disease severity such as MELD score? In addition, did the authors confirm longitudinal change in the titers of PR3-ANCA in each individual? Was a significant decrease in the titer found after liver transplantation?

Reply 2.3: Patient characteristics and laboratory values were determined at baseline (enrolment into the study). Patients were subsequently followed up for disease progression. In our study design, we did not collect consecutive samples.

To address the correlation of PR3-ANCA levels with disease severity scores, we added the following sentence to the Results/Association of PR3-ANCA … :

“PR3-ANCA levels were significantly correlated with MELD and extended MELD-Na (Spearman’s rho 0.194, 95% CI: 0.082 – 0.301, p < 0.001; 0.186, 95% CI: 0.074 – 0.294, p = 0.001, respectively).”

2.4. Previous studies that p-ANCA was commonly detected in patients with PSC should be cited in “Introduction”. It has been well established that PR3 is a main target antigen of c-ANCA. Did authors check immunofluorescent patterns of ANCA by the indirect immunofluorescence (p-ANCA or c-ANCA) in the enrolled patients seropositive for PR3-ANCA?

Reply 2.4: We thank this Reviewer for the helpful comment. We did not detect c-ANCA and p-ANCA in our study.  In another study, our collaborators found only a 55% concordance of RP3-ANCA with p- as well as c-ANCA and assumed a higher sensitivity of PR3-ANCA detection methods in contrast to indirect immunofluorescence (Laass et al., 2022 JPGN). We added the following sentence to the Introduction section:

“Perinuclear anti-neutrophil cytoplasmic antibodies (p-ANCA) or more precisely peripheral anti-neutrophil nuclear autoantibodies (p-ANNA) detected by qualitative in-direct immunofluorescence have been one of the most debated diagnostic markers in PSC in the past […]. However, p-ANNA do not appear to be directly linked to the clinical symptoms of the disease, and despite recent progress in identifying potential targets, their actual targets remain elusive […]. The involvement of autoimmunity in PSC is further strengthened by recent data demonstrating that ANCA to serine proteinase 3 (PR3-ANCA) detected by bead-based immunoassays can occur in PSC and have even predictive value for severe disease and occurrence of cholangiocarcinoma (CCA) [6–8].”

2.5. The result of the correlation between PR3-ANCA positivity and health-related QOL should be addressed in Table 4, although its statement was found in the text (lines 218-220). Moreover, the authors should describe or speculate the reason why PR3-positivity was not correlated with the QOL in patients with PSC, although it was significantly correlated with disease severity.

Reply 2.5: We thank this Reviewer for the helpful comment and have included the data in Table 4. We added the following sentences to the Discussion section:

“In our study, we confirmed the fact that PR3-ANCA are not linked with patient’s quality of life that is one of the most valid aspects of disease severity from a patient’s perspective [6]. Thus, unlike antimicrobial glycoprotein 2 (GP2), loss of tolerance to neutrophil targets does not appear to be associated with quality of life. Therefore, autoAbs to the large isoform of GP2 remain the only autoAbs to date that are linked with HRQoL enabling to target patient populations with a special need of attention. Furthermore, this finding appears to highlight the role of immunomodulators such as GP2 in the pathophysiology of PSC.

2.6. Solid triangles indicated individuals with high titer od PR3-ANCA in Figure 1. However, the reason why such patients were listed up from three categories (PSC, PBC and BD groups) remains uncertain. The authors should explain the reason for listing up such patients. Are there any clinical characteristics in such patients?

Reply 2.6: The presentation of samples with values higher than the upper quartile plus three times the interquartile range was based on statistical evaluation and was used to indicate samples with far out values. This is a common statistical approach to indicate extreme values in a cohort. PSC Patients with far out PR3-ANCA levels did not demonstrated different clinical characteristics but were younger at diagnosis and study entry.

We added the following sentence to the Results/Correlation … section:

“However, PSC patients with far out values (Fig. 1) were younger at diagnosis and study enrolment (24 years vs 29 years, 26 years vs 32 years, p < 0.05, respectively).”

2.7. The lower cut-off value (7.5U/ml) for PR3-ANCA was also set for the distinction of PSC patients from other patients. The setting of lower cut-off value led to the increase in sensitivity and decrease in specificity. Which is better as the cut-off value for PR3-ANCA, original one (15 U/ml) or lower one (7.5 U/ml)? The authors should answer the question.

Reply 2.7: We thank this Reviewer for the helpful comment. We added the following sentence to the Discussion section:

“Thus, depending on the clinical situation and an established PSC diagnosis, the lower cut-off of 7.5 U/mL may contribute to identifying more PSC patients at risk for severe disease.”  

2.8. Positive and negative predictive values (PPV and NPV) for PSC by PR3-ANCA detection should be also noted in Table 2.

Reply 2.8: We added the PPV and NPV values to Table 2.

2.9. The authors should describe the subtype of immunoglobulin in PR3-ANCA positive patients.(IgA dominant? Or IgG dominant?) In addition, what did AIH features mean in Table 3? The authors should clearly describe the definition of autoimmune features. Serum IgG levels and the prevalence of ANA should be determined in PSC patients.

Reply 2.9: We have changed “AIH features” in Table 3 to “AIH overlap” and throughout the text. This clinical characteristic was determined in patients in accordance with the latest EASL guidelines (EASL J Hepatol 2022).

The detection of PR3-ANCA IgG has been described in the Methods/Detection of PR3-ANCA section. To highlight the detection of IgG to PR3 we changed the following sentence in this section:

“PR3-ANCA IgG levels were ascertained by a novel third-generation ELISA (anti-PR3hs, GA Generic Assays GmbH, Dahlewitz, Germany) employing purified native PR3 as solid-phase antigen.”

Minor

2.10. Table 2 (line 231) should be corrected to Table 4.

Reply 2.10: We corrected the typo in the text.

2.11. INR (line 226) should be corrected to PT-INR.

Reply 2.11: We corrected the abbreviation in the text.

Reviewer 3 Report

The authors describe the correlation between the severity of the disease in PSC and the detection of PR3-ANCA by a novel third-generation ELISA.

In PSC patients with concurrent IBD, particularly those with UC, PR3-ANCA positivity was significantly more common and was accompanied by substantially higher bilirubin and liver enzyme levels. Additionally, a strong correlation between PR3-ANCA positivity and shorter LTx-free survival was found.

The methodology used in this well-written paper is appropriate, and the findings add some fresh, intriguing information to what we already know about the subject. I propose briefly discussing the microbiota's role in the context of patients who are PR3-ANCA positive. If the authors have not assessed the microbiota, they should include this information in the study limitations.

Author Response

We thank the Reviewer for her/his favorable comments. We would like to reply as follows:

R1: The methodology used in this well-written paper is appropriate, and the findings add some fresh, intriguing information to what we already know about the subject. I propose briefly discussing the microbiota's role in the context of patients who are PR3-ANCA positive. If the authors have not assessed the microbiota, they should include this information in the study limitations.

Reply: To adress the helpful comment we added the following sentences to the Discussion section:

"Neutrophils, as innate immune cells that can interact with the microbiota, infiltrate bile ducts of patients with PSC [35]. The occurrence of PR3-ANCA in PSC could be the result of impaired interaction of neutrophils with the biliary or intestinal microbiota."

"Furthermore, no data on the composition of the microbiota correlating with PR3-ANCA levels were obtained. " 

Round 2

Reviewer 1 Report

No international, current guidelines neither Expert opinion papers report the presence and diagnostic value of PR3-ANCA in PSC patients who classically develop the "atypical ANCA (pANNA)" targeting a different and well-characterized autoantigen. All relevant literature on ANCA in PSC should be discussed and compared.

Author Response

We thank the Reviwer for her/his comment.

R: No international, current guidelines neither Expert opinion papers report the presence and diagnostic value of PR3-ANCA in PSC patients who classically develop the "atypical ANCA (pANNA)" targeting a different and well-characterized autoantigen. All relevant literature on ANCA in PSC should be discussed and compared.

Reply: Tornai et al. have reviewed serological biomarkers for the management of primary sclerosing cholangitis recently (World J Gastroenterol 2022;28(21): 2291-2301. The authors discussed the value of atypical ANCA and PR3-ANCA for the serological diagnosis of PSC.

To the best of our knowledge, we discussed the available literature on ANCA and cross-referenced it with the new findings of our study.